# Optimized Detoxification of a Live Attenuated Vaccine Strain (SG9R) to Improve Vaccine Strategy against Fowl Typhoid

**DOI:** 10.3390/vaccines9020122

**Published:** 2021-02-03

**Authors:** Nam-Hyung Kim, Dae-Sung Ko, Eun-Jin Ha, Sunmin Ahn, Kang-Seuk Choi, Hyuk-Joon Kwon

**Affiliations:** 1Laboratory of Poultry Medicine, College of Veterinary Medicine, Seoul National University, Seoul 08826, Korea; hyacinthus@snu.ac.kr (N.-H.K.); kdskos1@snu.ac.kr (D.-S.K.); flower678@snu.ac.kr (E.-J.H.); vicky.ahn@snu.ac.kr (S.A.); 2Research Institute for Veterinary Science, College of Veterinary Medicine, BK21 for Veterinary Science, Seoul 08826, Korea; 3Laboratory of Avian Diseases, College of Veterinary Medicine, Seoul National University, Seoul 08826, Korea; 4Farm Animal Clinical Training and Research Center (FACTRC), GBST, Seoul National University, Seoul 08826, Korea

**Keywords:** *Salmonella enterica* serovar Gallinarum biovar Gallinarum, SG9R, detoxification, lipid A, vaccines, pro-inflammatory cytokines

## Abstract

The live attenuated vaccine strain, SG9R, has been used against fowl typhoid worldwide, but it can revert to the pathogenic smooth strain owing to single nucleotide changes such as nonsense mutations in the *rfaJ* gene. As SG9R possesses an intact *Salmonella* plasmid with virulence genes, it exhibits dormant pathogenicity and can cause fowl typhoid in young chicks and stressed or immunocompromised brown egg-laying hens. To tackle these issues, we knocked out the *rfaJ* gene of SG9R (named Safe-9R) to eliminate the reversion risk and generated detoxified strains of Safe-9R by knocking out *lpxL*, *lpxM*, *pagP*, and *phoP/phoQ* genes to attenuate the virulence. Among the knockout strains, live Δ*lpxL*- (Dtx-9RL) and Δ*lpxM*-9R (Dtx-9RM) strains induced remarkably less expression of inflammatory cytokines in chicken macrophage cells, and oil emulsion (OE) Dtx-9RL did not cause body weight loss in chicks. Live Dtx-9RM exhibited efficacy against field strain challenge in one week without any bacterial re-isolation, while the un-detoxified strains showed the development of severe liver lesions and re-isolation of challenged strains. Thus, SG9R was optimally detoxified by knockout of *lpxL* and *lpxM*, and Dtx-9RL and Dtx-9RM might be applicable as OE and live vaccines, respectively, to prevent fowl typhoid irrespective of the age of chickens.

## 1. Introduction

SG9R is a live attenuated vaccine strain that has long been inoculated subcutaneously to provide protection against fowl typhoid (FT) caused by *Salmonella enterica* serovar Gallinarum biovar Gallinarum (SG) and the food poisoning risk posed by *S. enterica* serovar Enteritidis (SE) infection in egg-laying hens in several countries [1,2]. SG9R was transformed into a rough strain by a nonsense mutation in the *rfaJ* gene [3]. However, the fact that the attenuation is attributable to only a single point mutation continues to raise concerns regarding pathogenic reversion. Previous studies have demonstrated that the field-isolated SG from SG9R inoculated farms exhibits the same DNA fingerprint as SG9R, and several SG9R-like rough strains have been reported [3,4,5]. SG9R not only has the potential to restore pathogenicity, but also has residual pathogenicity of its own. It can be transmitted vertically, and mortality and re-isolation can occur if there is immune suppression by insufficient nutrition or infection by immunosuppressive pathogens [3,6]. As SG9R inoculation decreases the growth rate in young chicks, SG9R vaccination has usually been recommended after 6 weeks of age, resulting in the lack of implementation of proper measures to protect against FT during the most susceptible period [7,8].

Administration of the rough vaccine strain SR2-N6 defective in *spvB* and *spvC* genes did not cause mortality or lesions even after fasting; thus, the residual pathogenicity of SG9R might be related to the intact *spv* locus of a large *Salmonella* virulence plasmid [9]. Presence of the *Salmonella* endotoxin decreases body weight in broiler chickens, and lipid A is at the core of the toxic moiety [10,11]. To detoxify lipid A, multiple enzymes involved in the modification of a lipid A precursor, lipid IV_A_, have been targeted in *Escherichia coli* and *Salmonella* serovar Typhimurium [12,13]. When the *lpxL* or *lpxM* gene is removed, the penta-acyl chain is generated, and the double mutant of *lpxL* and *lpxM* harbors tetra-acylated lipid A [13,14,15]. The penta-acylated lipopolysaccharide (LPS) of *lpxL* and *lpxM* mutants express reduced toxicity of LPS, resulting in the attenuation of virulence. These attenuated bacteria showed potential as vaccine candidates [16,17]. It is challenging to achieve an optimal attenuation of the SG vaccine, for induction of sufficient protection efficacy against the fatal field strain, and to circumvent mortality and persistent infection in young chicks and immunocompromised chickens.

In this study, we developed Safe-9R by knocking out the *rfaJ* gene of SG9R and examined the efficacy and toxicity of its live and killed vaccines. Furthermore, we generated mutant strains of Safe-9R by knocking out *lpxL*, *lpxM*, *pagP*, and *phoP/phoQ* (*phoP/Q*) genes, which are involved in lipid A biosynthesis. The toxicity was determined by analyzing the expression level of pro-inflammatory cytokines in vitro and by comparing the effects of the modified vaccines on body weight loss in young chicks. Additionally, pathogenicity and protective efficacy were assessed in chickens.

## 2. Materials and Methods

### 2.1. Bacteria and Experimental Birds

A field strain, SNU5161, showing characteristics similar to those of SG9R, was used to generate knockout mutant strains. A virulent field strain, SG0197, was used to test the efficacy of the vaccines [9]. The bacteria were cultured in Luria Bertani (LB) broth (Duchefa Biochemie, Groot Bijgaarden, Belgium) by incubating under shaking conditions (250 rpm) at 37 °C overnight aerobically.

One day-old male Hy-Line brown layer chicks without SG vaccination were obtained from a farm (Yangji Farm, Pyeongtaek-si, Republic of Korea), and had ad libitum access to feed and water. All animal experiments were approved by the Institutional Animal Care and Use Committee of BioPOA Co. (Hwaseong-si, Korea) (permission number BP-2020-001-1).

### 2.2. Generation of Knockout Mutant Strains

Safe-9R and Safe-9R-derived knock-out strains were constructed using the Red/ET recombination kit (Gene Bridges, Heidelberg, Germany) according to the manufacturer’s instructions [18]. Briefly, for homologous recombination, the homology arms were amplified via PCR by attaching oligonucleotides to target the genes (Appendix A). The Red/ET plasmid and amplified homology arms were transformed to SG9R or Safe-9R using an GenePulser Xcell (Bio-Rad, Hercules, CA, USA) at 2500 V, 10 μF, and 600 Ω with 1 mm slit Gene Pulser cuvettes (Bio-Rad). Transformed bacteria were selected using LB agar with tetracycline or kanamycin.

### 2.3. PCR and Sequencing

All generated mutants were confirmed via PCR and sequencing. Bacterial genomic DNA was extracted using the G-spin Genomic DNA Extraction Kit for Bacteria (iNtRON Biotechnology, Seongnam-si, Korea) and PCR was conducted under the following conditions: 1 μL of the template DNA (50 ng/μL), 3 μL of 10 × buffer, 3 μL of dNTPs (5 mM), 0.5 μL of each primer (10 pmol/μL), and 0.25 μL of Taq polymerase (MGmed, Seoul, Korea). The thermal cycling conditions were as follows: 95 °C for 5 min, followed by 35 cycles at 95 °C for 30 s, 55 °C for 30 s, 72 °C for 1 min, and 72 °C for 5 min. PCR amplicons were purified using a PCR/Gel Purification Kit (MGmed), and sequencing was performed using the ABI 3711 automatic sequencer (Cosmogenetech, Seoul, Korea). Nucleotide sequences were translated and compared using the BioEdit program version 7.2.5. Primers used in this study are described in Appendix A.

Safe-9R and SG9R were also subjected to next-generation sequencing, which was performed as per methods described previously [19]. Briefly, the extracted genomic DNA was sequenced using the HiSeq 2000 platform (Illumina, San Diego, CA, USA) and the filtered data were mapped using BWA version 0.7.12 to *S. enterica* serovar Gallinarum str. 287/91 (GenBank Accession Number NC_011274.1) in the National Center for Biotechnology Information database. The genes that differed between Safe-9R and SG9R were identified.

### 2.4. Detection of Pro-Inflammatory Cytokine Expression in HD11 Cells

The chicken macrophage cell line, HD11, was cultured using the Roswell Park Memorial Institute Medium (RPMI) 1640 medium (Thermo Fisher Scientific, Waltham, MA, USA) modified with L-glutamine and phenol red and supplemented with 10% fetal bovine serum (FBS; Thermo Fisher Scientific). Two days before the *Salmonella* infection, aliquots of the HD11 cell suspension (1 × 10^6^ cells/mL) were seeded into each well of a 24-well plate (SPL Life Sciences, Pocheon-si, Korea) at a volume of 500 μL/well and cells were allowed to grow to approximately 85% confluence.

The overnight cultured bacteria were adjusted to an optical density of 0.2 at 600 nm and a ten-fold dilution was performed using phosphate-buffered saline (PBS) to obtain multiplicity of infection of 10. The diluted bacterial suspension was centrifuged at 11,000 g for 1 min and resuspended in RPMI 1640. The bacterial suspensions were inoculated after washing the cells twice with the medium and incubated for 2 h at 37 °C in a 5% CO_2_ incubator. At 2 h post-infection, the cells were washed once and incubated with 150 μL/mL of gentamicin sulfate (Duchefa Biochemie)-supplemented medium for another 2 h. After washing the cells with the medium, total RNA was extracted using the RNeasy Mini Kit (Qiagen, Hilden, Germany). cDNA was synthesized from equal amounts of total RNA using the amfiRivert cDNA Synthesis Platinum Master Mix (GenDEPOT, Katy, TX, USA).

Quantitative PCR (qPCR) analysis was performed to compare the expression of pro-inflammatory cytokines (interleukin (IL)-1β, tumor necrosis factor (TNF)-α, and IL-18), LPS (lipopolysaccharide) -recognizing receptor (Toll-like receptor-4 (TLR-4)), and inducible nitric oxide synthase (iNOS). Briefly, 10-μL reaction mixtures contained 5 μL of 2X AMPIGENE qPCR Green Mix Hi-ROX (Enzo Life Sciences, Farmingdale, NY, USA), 0.5 μL each of the forward and reverse primers, and 1 μL of cDNA. The normalization was performed using glyceraldehyde 3-phosphate dehydrogenase. Primers used in qPCR are listed in Appendix A [20,21,22]. The mRNA expression levels of each pro-inflammatory cytokine were compared using the 2 ^–∆∆*C*t^ method.

### 2.5. Growth Temperature Sensitivity Test

The detoxified strains Δ*lpxL* and Δ*lpxM* were evaluated for proliferation at 42 °C using Safe-9R as a control. Each vaccine strain (1 × 10^2^cfu/plate) was spread on MacConkey agar and checked for growth for 2 days.

### 2.6. Analysis of Protective Efficacy and Toxicity of Safe-9R

Live Safe-9R was inoculated at 6 and 18 weeks at 1 × 10^7^ colony forming units (cfu)/chicken, and 4 weeks after the second vaccination, the pathogenic field strain SG0197 (1 × 10^8^ cfu/chicken) was challenged per os.

The oil emulsion (OE) Safe-9R vaccine was prepared by heat inactivation of Safe-9R at 65 °C for 2 h in a water bath, followed by gradual cooling to room temperature and emulsification of bacteria and oil adjuvant (Montanide ISA 70, Seppic, La Garenne-Colombes, France) at a ratio of 3:7. OE Safe-9R was administered via the intramuscular route to 1-week old brown Hy-Line chicks (approximately 1 × 10^9^ cfu/100 μL/chick). SG0197 was challenged in vaccinated and negative control groups at 2 and 7 week-post-vaccination (wpv), respectively, and the serum samples were collected before the challenge. Mortality was observed for 2 weeks, and the chickens were subjected to fasting conditions for three days to detect persistently infected chickens using the protein–energy malnutrition (PEM) model [3].

### 2.7. Assessment of Humoral Immunity and Weight Gain after Administration of OE Vaccines of Detoxified Strains

The OE vaccines of detoxified strains were prepared as per the methods described for OE Safe-9R. They were diluted to 3 × 10^9^ cfu/100 μL and mixed at a 3:7 ratio with the oil adjuvant Montanide ISA 70. Mixed vaccines were administered via the intramuscular route to fifteen 1 week-old brown Hy-Line chicks in each group, which were divided into Δ*lpxL*, Δ*lpxM*, Safe-9R, SG9R, and negative control. Blood samples were collected 2 weeks after vaccination. Antibody titers of OE vaccines were evaluated using OmpA and OmpX peptide ELISA, and outer membrane protein (OMP) extract ELISA, as per the previously described methods [8]. OE vaccines were inoculated into 1 and 2 week-old chicks, and the body weight was measured weekly for 2 weeks after vaccination.

### 2.8. Evaluation of Protection Efficacy and Pathogenicity of Live Detoxified Vaccines

To test the protective efficacy of detoxified strains, 1 day-old (d-o) and 1 week-old (w-o) chicks (10 chicks/group) were assigned to the same scheme of groups as above experiment of the OE detoxified vaccines. Live vaccine strains were diluted to 1 × 10^7^ cfu/100 μL and were administered via the subcutaneous route. At 2 w-o, the chicks were infected orally with the pathogenic field strain SG0197 (1 × 10^6^ cfu/chick). To test the pathogenicity of the detoxified strains, five 1 week-old chicks were assigned to the same scheme of groups and inoculated with the same dose as above. After 2 weeks they were subjected to PEM conditions. Re-isolation of bacteria was performed by scrubbing a sterile cut section of the liver on MacConkey agar (Becton Dickinson, Franklin Lakes, NJ, USA). Isolated bacteria were identified by plate agglutination test to distinguish rough vaccine from smooth challenge strains. For plate agglutination test, isolated colony was suspended in 30 μL sterile saline, and 5 μL of *Salmonella* D group O antibody factor 9 (Becton Dickinson) was added and mixed by tilting for 30 s. If agglutination occurred, the isolates were determined to be the smooth strain. IgA detection in bile juice was performed as per the previously described method [8].

### 2.9. Assessment of T-Cell Stimulation by Fluorescence Activated Cell Sorting

The detoxified live vaccines were administered to 1 day-old brown Hy-Line chicks (10 per group; as mentioned above). Whole blood samples were collected in heparin-containing tubes and the same volume was pooled by group. Peripheral blood mononuclear cells (PBMCs) were isolated using Lymphoprep (Axis Shield, Dundee, Scotland) and washed with PBS supplemented with 2% FBS (Thermo Fisher Scientific). The number of PBMCs was counted and adjusted to a density of 10^6^ cells/mL. Three-microliter volume of CD8^+^ T-cell antibody-fluorescein isothiocyanate (LSBio, Seattle, WA, USA) and CD4 T-cell antibody-allophycocyanin (SouthernBio, Bermingham, AL, USA) was inoculated into fifty-microliter aliquot of cells from each group, and another fifty-microliter aliquot was prepared as a control. All aliquots were incubated for 15 min on ice in the dark. After incubation, the cells were washed and resuspended in 300 μL of PBS. Samples were analyzed using FACSCalibur (Becton Dickinson).

### 2.10. Statistical Analysis

Statistical analyses were performed using SPSS Statistics version 26.0 (IBM, Chicago, IL, USA). One-way analysis of variance was used to analyze the significant differences between the groups along with the Bonferroni post-hoc test. The significance of the group that followed the normality but violated equal variance was confirmed using the Games–Howell test. Data that did not follow a normal distribution were analyzed using the Kruskal–Wallis *H* test, and the Bonferroni correction was used as the post-hoc test. *p* value less than 0.05 was considered statistically significant.

## 3. Results

### 3.1. Generation of Safe-9R

We knocked out the *rfaJ* gene of SG9R and confirmed the amplicons by PCR (Figure 1A). We named the *rfaJ* knockout strain Safe-9R and compared its genome sequences with those of SG9R by re-sequencing. There were no differences except for the *rfaJ* gene (data not shown).

### 3.2. Generation and Characterization of Gene Knockout Strains for Detoxification

We knocked out target genes (*phoP*/*Q*, *lpxL*, *lpxM*, and *pagP*) by replacing the antibiotic resistance gene and the amplicons of target genes of Safe-9R; as a result, the knockout strains were the same in size (Figure 1B). The toxic effects of the knockout strains were compared using reverse transcription quantitative PCR of *IL-1β*, *IL-18*, *TNF-α*, *iNOS*, and *TLR-4* mRNAs. The mRNA levels of *TNF-α* were not significantly different between the knockout strain-infected HD-11 cells and uninfected HD-11 cells. However, the *IL-1β*, *IL-18*, and *iNOS* mRNA levels of *phoP/Q-* and *pagP*-knockout strains, and Safe-9R were significantly higher than those of *lpxL*- and *lpxM*-knockout strains (named Dtx9RL and Dtx-9RM, respectively), and the negative control. Only Safe-9R showed significantly higher *TLR-4* mRNA levels than the negative control (Figure 2). Therefore, live Dtx9RL and Dtx-9RM were successfully detoxified and did not induce the transcription of *IL-1β*, *IL-18*, *iNOS*, and *TLR-4* genes.

In contrast to Safe-9R, Dtx-9RL and Dtx-9RM showed poor proliferation on the surface of MacConkey agar at 42 °C. In contrast to Safe-9R colonies, Dtx-9RL and Dtx-9RM were barely visible on the first day of incubation but were visible on the second day of incubation.

### 3.3. Protective Efficacy and Humoral Immunogenicity of Safe-9R

Live Safe-9R vaccines were markedly protective against the fatal challenge of the field strain, resulting in no mortality in the vaccinated group, in contrast to 100% mortality in the unvaccinated group (Table 1). Although the survival rate of one out of the three experiments confirmed significance, the OE Safe-9R vaccination in 1 week-old chicks protected against mortality in fatal challenge at 2 wpv. However, the OE Safe-9R vaccination did not protect from mortality at 7 wpv and even showed lower survival rates in vaccinated than unvaccinated groups in two (50 versus 90 and 70 versus 90) out of the three experiments (Table 1). Interestingly, anti-OmpA and anti-OmpX antibody levels in the vaccinated group at 2 wpv were significantly higher than those of the unvaccinated group, similar to the anti-OMP antibody levels (Figure 3). However, only the anti-OMP antibody level in the vaccinated group at 7 wpv was significantly higher than that of the negative control group (Figure 3).

### 3.4. In Vivo Verification of Detoxification

OE vaccines Dtx-9RL, Dtx-9RM, and Safe-9R were administered to 2 w-o chicks and their body weights were measured every week for 2 weeks. Only Safe-9R-vaccinated chicks showed significantly lower body weight than the negative control (*p* < 0.05; Figure 4B). To differentiate body weight changes between detoxified strains, we performed the same experiment using 1 w-o chicks and including SG9R. The body weight of Dtx-9RL-vaccinated chicks was not significantly different from that of the negative control, but body weights of chicks vaccinated with Dtx-9RM, Safe-9R, and SG9R were significantly less than those of the negative control. Therefore, detoxification of Dtx-9RL was apparent in the 1 w-o chick body weight model (Figure 4A).

### 3.5. Humoral Immunogenicity of OE Vaccines of Detoxified Strains

The humoral immunogenicity of OE vaccines Dtx-9RL and Dtx-9RM was compared with that of Safe-9R and SG9R in 1 w-o chicks. The anti-OmpA, anti-OmpX, and anti-OMP antibody levels in all the vaccinated groups were significantly higher than those in the negative control group at 2 wpv and OE vaccines of the detoxified strains exhibited similar immunogenicity to Safe-9R and SG9R (Figure 5).

### 3.6. Evaluation of Residual Pathogenicity and Protective Efficacy of Live Detoxified Strains

The residual pathogenicity of live detoxified and un-detoxified strains was compared in the PEM model. No lesions or mild lesions in the liver were observed with the administration of Dtx-9RL and Dtx-9RM, respectively, and they were negative for bacterial re-isolation. However, Safe-9R and SG9R caused moderate to severe lesions and were positive for bacterial re-isolation (4/5 and 3/5), respectively (Table 2).

The protective efficacies of live detoxified strains were evaluated by vaccination of 1 d-o chicks and 1 w-o chicks. The field strain was challenged at 2 w-o, causing 10 and 40% mortality in the Dtx-9RL-vaccinated group, while the negative control group showed 20 and 30% of mortality, respectively. The Dtx-9RM-, Safe-9R- and SG9R-vaccinated groups showed no mortality on both challenges. When autopsied, chickens in the Dtx-9RM-vaccinated group showed much less severe lesions in the livers than the other groups. However, Safe-9R and SG9R-vaccinated and the negative control groups showed more moderate to severe liver lesions than the Dtx-9RM-vaccinated group (Table 3). The grading of the liver lesion is demonstrated in Appendix A. When the field strain was challenged at 1 wpv, bacteria were re-isolated from all the groups except Dtx-9RM-vaccinated group and they were identified to be the smooth challenge strain. When challenged at 2 wpv, only the rough vaccine strain was re-isolated from the SG9R-vaccinated group (Table 4).

The humoral and mucosal antibody levels were compared, and Dtx-9RM-, Safe-9R-, and SG9R- vaccinated groups showed significantly higher levels than those of the negative control, but Dtx-9RL-vaccinated groups did not exhibit such levels (Figure 6). Dtx-9RL- and Dtx-9RM-vaccinated groups showed a significantly higher percentage of CD8^+^ T cells in PBMCs collected at 1 wpv compared to those observed in the negative control. Safe-9R-vaccinated, and Dtx-9RM- and SG9R-vaccinated groups showed significantly higher percentages of CD8^+^ T cells than those of the negative group at 1week post-challenge (wpc) and 2 wpc, respectively (Figure 7A). All of the vaccinated groups, and SG9R- and Dtx-9RM-vaccinated groups showed a significantly higher percentage of CD4^+^ T cells than the negative control group at 1 wpc and 2 wpc, respectively (Figure 7B).

## 4. Discussion

SG9R and SR2-N6 have been used to provide protection in chickens against FT, and paratyphoid caused by SE [9,23]. The underlying reason for SG vaccines showing cross-protective efficacy to SE may be attributed to the intimate genetic relationship between SG and SE, and the competitive exclusion of SE by SG [24,25]. Since SG9R was permitted for use in commercial layer farms in 2001, food poisoning cases caused by SE have decreased gradually in Korea. In broiler chickens which are not permitted for vaccination, frequent FT outbreaks and SE isolation have been reported [26,27]. In the EU, SG9R was used to reduce food poisoning cases caused by SE, until FT outbreaks were reported in SG9R-vaccinated layer farms [5]. Although field pathogenic isolates originating from SG9R due to a single point mutation in *rfaJ* has never been reported, SG9R has been shown to cause FT in immunocompromised flocks [3,28]. Therefore, removal of this reversion risk and maintenance of protective efficacy without residual pathogenicity are crucial for successful SG9R vaccination.

In this study, we eliminated the reversion risk of SG9R by knocking out *rfaJ* and demonstrated that the live Safe-9R vaccine was as efficacious as SG9R. OE Safe-9R induced production of significantly high titers of specific antibodies to linear epitopes of OmpA and OmpX, and OMPs at 2 wpv, as previously reported [8]. However, these specific antibodies disappeared at 7 wpv (Figure 3). The correlation of higher anti-OmpA and anti-OmpX antibodies with protective efficacy may highlight their immunoprotective roles. Even though significantly high anti-OMP antibody levels were apparent at 7 wpv, this observation did not highlight efficacy or showed even worse outcomes than the control group. This observation may be reminiscent of antibody-dependent enhancement (ADE), which is defined as the suppression of host defense by the immune complex composed of specific IgG and antigen, and the possibility of ADE in *Salmonella* has been reported [29]. Therefore, further studies on the protective efficacy of subunit vaccines composed of OmpA and OmpX epitopes and the relation of OMP antibodies against ADE should be conducted. The negative effect of OE Safe-9R on the body weight of young chicks necessitates the development of increased detoxified vaccine strains.

The detoxification of Safe-9R in terms of the reduced induction of pro-inflammatory cytokines was apparent when *lpxL* or *lpxM* was knocked out compared to *pagP* and *phoP/Q* in an in vitro model (Figure 2). The detoxification levels of Dtx-9RL and Dtx-9RM and body weight of 2 week-old chicks were indistinguishable in the in vitro model, but they were clearly differentiated by the body weight in 1 week-old chicks (Figure 4). To date, endotoxins are known to cause reductions in the body weight of chickens, and we demonstrated that knockout of *lpxL* was the best strategy to eliminate these side effects [7].

In the PEM model, Dtx-9RL was also the least pathogenic, showing no lesions or re-isolation in the liver of inoculated chicks. Dtx-9RM showed mild lesions without re-isolation of bacteria, but Safe-9R and SG9R showed moderate to severe lesions with re-isolation of the challenged strain from the liver. Therefore, different levels of pathogenicity among the detoxified strains were also clearly differentiated by the PEM model. Additionally, the significantly higher antibodies in the group administered with OE Dtx-9RL compared to the control group may support the use of Dtx-9RL for growing young chicks and laying hens that are sensitive to endotoxin.

The protective efficacies of live vaccines of Dtx-9RM were verified in our radical protection model. When challenged at 1 wpv, Safe-9R- and SG9R-vaccinated chickens showed severe lesions after field strain challenge, and re-isolated bacteria were identified as the challenged field strains (Table 4). This was an unexpected result and might be attributed to early infection by the field strain after the vaccination. Our previous study demonstrated that competition between field strains with different pathogenicity might result in the dominant isolation of more pathogenic strains over time [19]. Therefore, the pathogenicity of Safe-9R and SG9R may not be overcome by the host immunity within 1 wpv and subsequent challenge may result in the development of more severe lesions and predominant persistent infection by more pathogenic challenge strains. Considering no re-isolation of challenged smooth strains from Safe-9R- and SG9R-vaccinated groups when challenged at 2 wpv, they may require a longer length of time for full protection. However, the protective efficacy of live Dtx-9RM was demonstrated in terms of the bacterial re-isolation rate and the severity of lesions, and the priming of mucosal immunity by Dtx-9RM—similar to that observed with Safe-9R and SG9R—was also demonstrated (Figure 6) [8]. Additionally, the better protective efficacy of live Dtx-9RM vaccine can be supported by the significantly higher CD8^+^ and CD4^+^ T-cell responses after vaccination and challenge (Figure 7). The protective efficacy of the live Dtx-9RL vaccine was not enough, showing similar mortality to the negative control and Dtx-9RL seemed to be over attenuated.

Both *lpxL* and *lpxM* are responsible for late secondary acylation, which provide lauroyl and myristoyl groups, respectively, to the different sites of the tetra-acylated intermediate, lipid IV_A_ [14,15]. Therefore, different chemical structures may be variably sensed by the innate and acquired immune systems of chickens. According to previous reports, in contrast to *lpxM*, defects in *lpxL* resulted in reduced phagocyte resistance and detoxified reaction in the *Limulus* amebocyte lysate test [12,30,31,32]. Therefore, the in vivo tests performed in this study may be useful strategies to differentiate *lpxL-* and *lpxM*-knockout mutants. The attenuation of Dtx-9RL and Dtx-9RM can also be explained by the reduced growth rate at the normal body temperature of chickens at 42 °C. The vulnerability of the *lpxL* mutant to high growth temperatures has already been reported in *E. coli* and *Salmonella* serovar Typhimurium, but we observed a similar effect of the *lpx*M-knockout mutation in SG [33,34].

## 5. Conclusions

In conclusion, we developed Safe-9R to eliminate the reversion risk of SG9R and generated Dtx-9RL and Dtx-9RM by detoxifying Safe-9R to optimally attenuate the virulence. Dtx-9RL was sufficiently detoxified to be used as OE vaccine, but further study was needed as it could not provide long-term protection. Dtx-9RM might be the superior substitute for live SG9R vaccine as it has lower virulence and could establish immunity more rapidly. Additionally, the developed strains might be useful for providing protection against paratyphoid caused by SE, as well as FT, in poultry.

## Figures and Tables

**Figure 1 vaccines-09-00122-f001:**
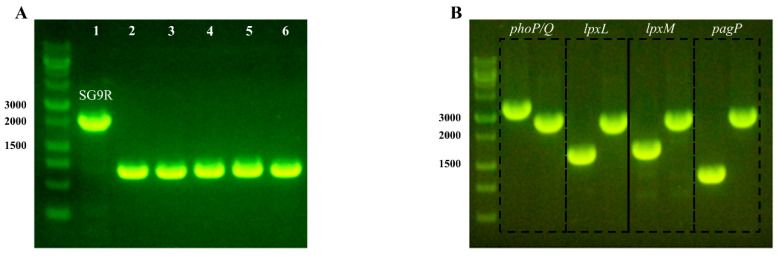
Deletion of *rfaJ* in Safe-9R and detoxified strains. (**A**) Deletion of *rfaJ* in Safe-9R and Safe-9R-derived knock-out strains. Lanes: 1, SG9R; 2, Safe-9R; 3, Δ*phoP/Q*; 4, Δ*lpxL*; 5, Δ*lpxM*; 6, Δ*pagP*. (**B**) Deletion of lipid A biosynthesis-related genes of Safe-9R-derived knock-out strains. The amplicon of Safe-9R for each gene was placed at the first lane of each rectangle to compare the amplicon of knock-out mutant strain.

**Figure 2 vaccines-09-00122-f002:**
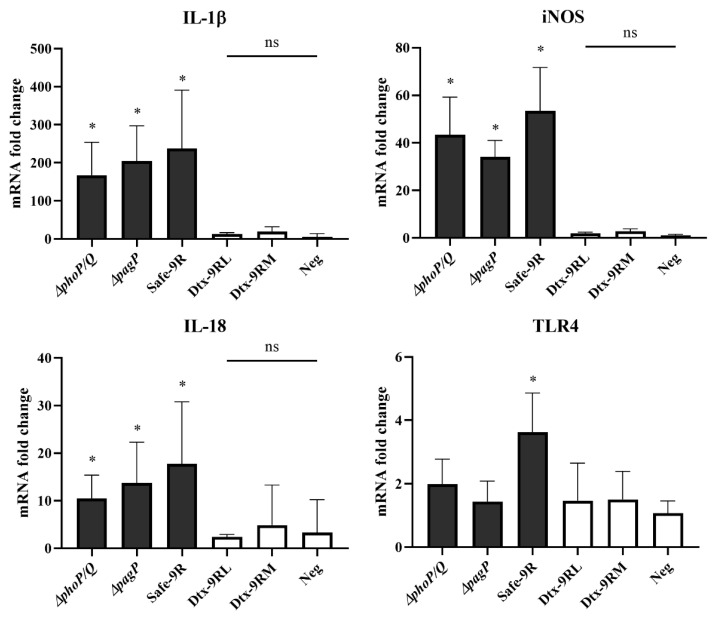
Comparison of stimulatory effects of knock-out mutant strains on transcriptions of pro-inflammatory cytokines and related genes in HD11 cells. Comparison of relative transcription levels of IL-1β, IL-18, iNOS and TLR4 genes in HD11 after infection of knock-out mutant strains was performed by using the 2–∆∆Ct method. Statistical significance indicated as follows: ns not significant, * significantly different compared with Neg (*p* < 0.05).

**Figure 3 vaccines-09-00122-f003:**
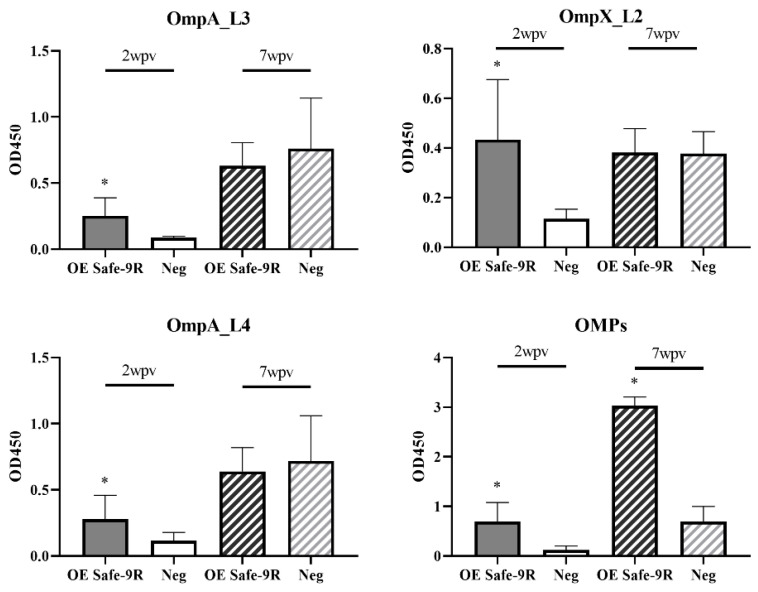
Humoral immune responses of the OE Safe-9R vaccines. Each group received OE Safe-9R at 1 week of age, and serum samples were collected at 2 weeks post-vaccination (wpv) and 7 wpv, respectively. The immune response was analyzed by ELISA made by *Salmonella enterica* serovar Gallinarum biovar Gallinarum (SG) immunogenic outermembrane proteins (OMP), OmpA and OmpX, and total OMP extracts. * Indicates a significant difference compared with Neg (*p* < 0.05).

**Figure 4 vaccines-09-00122-f004:**
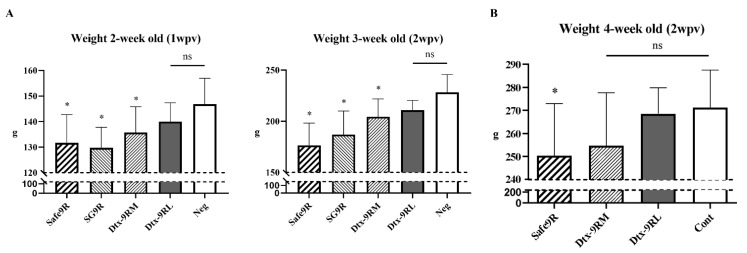
In vivo verification of detoxification by chick body weight model. Inactivated vaccines were inoculated in (**A**) 1 week-old and (**B**) 2 week-old chicks and the differences in body weight were examined. The significance was compared to the negative control. Statistical significance indicated as follows: ns not significant, * significantly different compared with Neg (*p* < 0.05).

**Figure 5 vaccines-09-00122-f005:**
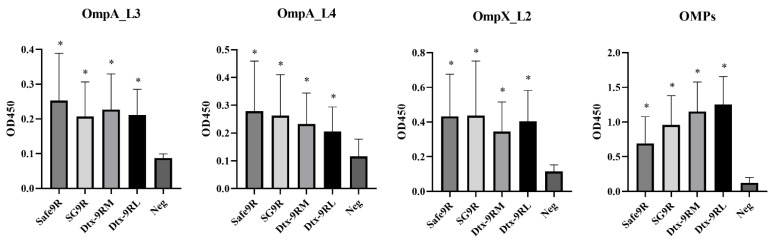
Humoral immunogenicity of oil emulsion vaccines of the detoxified strains. Vaccines were inoculated at 1 week of age, and antibody titers were determined after 2 weeks. * Indicates a significant difference compared with Neg (*p* < 0.05).

**Figure 6 vaccines-09-00122-f006:**
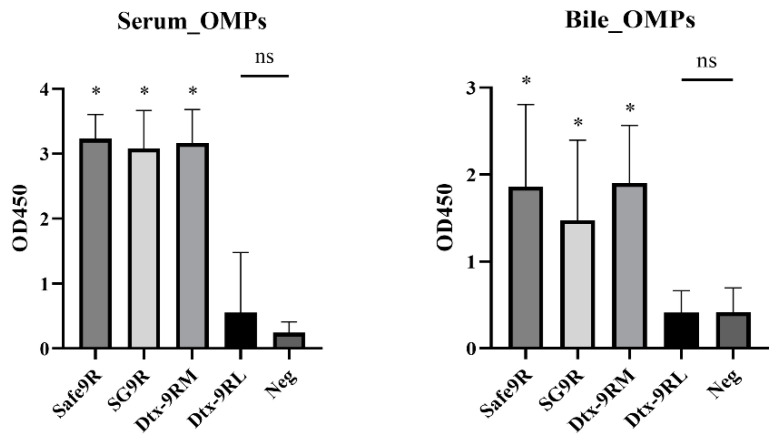
Humoral and mucosal immunogenicity of live vaccines with the detoxified strains after challenge. Detoxified vaccines were inoculated in 1 week-old chicks and the virulent field strain was challenged after 1 week of vaccination. Blood and bile samples were collected after 2 weeks of the challenge and analyzed with the ELISA. Statistical significance indicated as follows: ns not significant, * significantly different compared with Neg (*p* < 0.05).

**Figure 7 vaccines-09-00122-f007:**
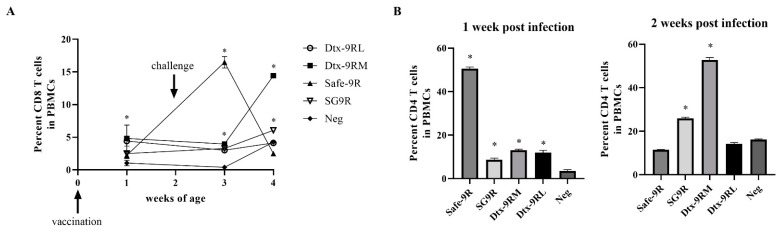
Proportion of CD8+ T cells in peripheral blood mononuclear cells (PBMCs) determined by fluorescence activated cell sorting. Detoxified vaccines were inoculated in 1 day-old chicks and the whole blood samples were collected in heparin-containing tubes. The samples were pooled by group and PBMCs were isolated. The percentage of (**A**) CD8+ T cells and (**B**) CD4+ T cells were analyzed. * Indicates a significant difference compared with Neg (*p* < 0.05).

**Table 1 vaccines-09-00122-t001:** Protective efficacy of live and oil emulsion vaccines of Safe-9R.

Group	Vaccination	Survival Rate ^b^ (%)		
Live Safe-9R	Vaccinated	100 (10/10) *		
	Non-vaccinated	0 (0/10)		
OE Safe-9R ^a^—2 wpv ^c^	Vaccinated	60 (6/10) *	50 (5/10)	80 (8/10)
	Non-vaccinated	0 (0/10)	11.1 (1/9)	50 (5/10)
OE Safe-9R ^a^—7 wpv ^c^	Vaccinated	87.5 (7/8)	50 (5/10)	70 (7/10)
	Non-vaccinated	80 (8/10)	90 (9/10)	90 (9/10)

^a^ Oil emulsion (OE) vaccine inoculated in 1 week-old and challenged in 3 week-old (2wpv) and 8 week-old (7wpv), respectively. OE Safe-9R efficacy tests were performed in triplicate. ^b^ Survival rate observed for 17 days, including the last 3 days in fasting condition. ^c^ wpv refers to week-post vaccination. * indicates a significant difference from the negative control (*p* < 0.05).

**Table 2 vaccines-09-00122-t002:** Re-isolation of detoxified strains in protein-energy malnutrition model *.

Groups	Dtx-9RL	Dtx-9RM	Safe-9R	SG9R	Negative
Lesion ^a^	0/5	5/5	5/5	5/5	0/5
Re-isolation ^b^	0/5	0/5	4/5	3/5	0/5

* fasted for three days after 2 weeks post-vaccination. ^a^ Presence of the liver lesions (No. of positive/No. of inoculated). ^b^ Recovery of the inoculated strains (No. of recovered/No. of inoculated).

**Table 3 vaccines-09-00122-t003:** Protective efficacy of live detoxified vaccines.

Group	Dtx-9RL		Dtx-9RM		Safe-9R		SG9R		Negative	
Vaccinated Age	A ^1^	B ^2^	A	B	A	B	A	B	A	B
0 ^a^	7	4	2	2	2	2	3	2	4	5
1 ^b^	1	0	8	4	0	1	1	1	0	1
2 ^c^	0	1	0	3	3	5	5	3	2	0
3 ^d^	1	1	0	1	5	2	1	4	2	1
4 ^e^	1	4	0	0	0	0	0	0	2	3
Severe liver lesions *	20%	60%	0%	40%	80%	70%	60%	70%	60%	40%
Number of chickens	10	10	10	10	10	10	10	10	10	10

Liver lesion scoring is demonstrated in Figure 1. ^a^ normal; ^b^ single to several necrotic foci (<5); ^c^ multiple necrotic foci (<100); ^d^ highly multiple necrotic foci (countless) and severe hepatomegaly; ^e^ death. ^1^ Vaccination at 1 week-old and challenged at 2 week-old (1wpv). ^2^ Vaccination at 1 day-old and challenged at 2 week-old (2 wpv). * Lesion score > 2.

**Table 4 vaccines-09-00122-t004:** Re-isolation and identification of bacteria after live detoxified vaccines inoculation and challenge.

Group	Challenged at 1 wpv ^a^					Challenged at 2 wpv				
Sample	Dtx-9RL	Dtx-9RM	Safe-9R	SG9R	Negative	Dtx-9RL	Dtx-9RM	Safe-9R	SG9R	Negative
Re-isolation	2/9 ^b^	0/10	4/10	3/10	4/8 ^b^	0/6 ^b^	0/10	0/10	1/10	0/7 ^b^
Smooth/Rough ^c^	10/0	Nt ^e^	10/0	5/0 ^d^	10/0	nt	nt	nt	0/4 ^d^	nt

^a^ wpv: week post-vaccination. ^b^ Bacteria re-isolation was not performed in deceased chickens. ^c^ Distinguished by plate agglutination test with commercial anti-O antigen antiserum. ^d^ Maximally 10 and all colonies were tested. ^e^ nt: not tested due to absence of colony.

## Data Availability

Data is contained within the article or supplementary material.

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
