# Peer review of "Optimized Detoxification of a Live Attenuated Vaccine Strain (SG9R) to Improve Vaccine Strategy against Fowl Typhoid"

_vaccines, 2021, doi:10.3390/vaccines9020122_

Round 1

Reviewer 1 Report

The manuscript number 1093211 entitled “Optimized detoxification of a live attenuated vaccine strain (SG9R) to improve vaccine strategy against fowl typhoid”. To attain this aim, authors knocked out the rfaJ gene of SG9R (named Safe-9R) to eliminate the reversion risk and generated detoxified strains of Safe-9R by knocking out lpxL, lpxM, pagP, and phoP/phoQ genes to attenuate the virulence. They also observed that Dtx-9RL and Dtx-9RM strains might be applicable as oil emulsion and live vaccines, respectively, to prevent fowl typhoid irrespective of the age of chickens. This work is well organized and present interesting results but some points should be improved before to be considered for publication.

  • In the beginning of Introduction should be more explicit the limitations of the actual vaccine based on the SG9R in order to make clear the need of improvement. What concerns and adverse events have been occurred?
  • A more comprehensive description of the actual vaccine should be included in the introduction, for instance, how is administered, with association of delivery vehicles, oil emulsions, adjuvants?
  • In section 2.1 should be completed the bacteria growth conditions. What are the shaking rotations? With presence our absence of O2? And so on…
  • In Table 2, Why it appears three results of survival rate for the last two groups? It should be included more specifications in the header of the table.
  • Also in description of results or in the discussion, should be explained why the survival rate of non-vaccinated is higher than the vaccinated of the group OE Safe-9Ra-7 wpvc?
  • In the end of discussion section, should be clarified what is the advantage of the present work? What is the practical application of this study? And What should be performed to approved the use of these optimal vaccine strains? Should be used several strains in combination? Or should be selected one? What is missing?

Author Response

  • In the beginning of Introduction should be more explicit the limitations of the actual vaccine based on the SG9R in order to make clear the need of improvement. What concerns and adverse events have been occurred?

- We agree to reviewer’s comment. We revised the sentences describing the limitation of the SG9R (page 1, lines 40 and 45)

  • A more comprehensive description of the actual vaccine should be included in the introduction, for instance, how is administered, with association of delivery vehicles, oil emulsions, adjuvants?

- We agree to reviewer’s comment. SG9R is a live attenuated vaccine (lyophilized form) and re-suspended in aqueous solution provided by manufacturer. Therefore, SG9R is a sole immunogen without any delivery vehicles or oil composition. For better understanding we revised as ‘SG9R is a live attenuated vaccine strain and has long been inoculated subcutaneously…’ in page 1, lines 33.

  • In section 2.1 should be completed the bacteria growth conditions. What are the shaking rotations? With presence our absence of O2? And so on…

- We agree to reviewer’s comment. We added the information of shaking rotation and aerobic condition (page 2, line 76).

  • In Table 2, Why it appears three results of survival rate for the last two groups? It should be included more specifications in the header of the table.

- The genetic background of Safe-9R is identical to SG9R except the way of attenuation of rfaJ gene function (large deletion vs nonsense mutation). The protection efficacy of live Safe-9R was verified as expected and we did not repeat the experiment. However, the efficacy of OE SG9R has never been investigated, and we repeated the experiment 3 times to ensure the efficacy of OE Safe-9R. We added the description in the footnote of the Table 1 (page 7, line 269).

  • Also in description of results or in the discussion, should be explained why the survival rate of non-vaccinated is higher than the vaccinated of the group OE Safe-9Ra-7 wpvc?

- At this moment we have not enough data to clearly explain the result, and that’s why we described possible involvement of antibody-dependent enhancement (ADE) which was reported in Salmonella infection (page 11, line 400-403). We are going to study in further in the near future.

  • In the end of discussion section, should be clarified what is the advantage of the present work? What is the practical application of this study? And What should be performed to approved the use of these optimal vaccine strains? Should be used several strains in combination? Or should be selected one? What is missing?

- We agree to reviewer’s comment. We revised the end of discussion section (page 12, line 454-460).

Reviewer 2 Report

Kim et al. described the detoxification of two live attenuated vaccine strains SG9R, specifically the Dtx-9RL and Dtx-9RM strains lacking lpxL or lpxM genes to attenuate virulence. The manuscript is very well written and are experimentally sound. In the introduction, the authors indicates that penta-acylated LPS of lpxL and lpxM mutants, but not double mutant, expresses reduced toxicity of LPS itself. I am wondering if authors had any additional data they can incorporate showing the effect of double mutant strain in the virulence?

Minor comment: Perhaps Table 1 can go into supplementary section to improve readability.

Author Response

  • Kim et al. described the detoxification of two live attenuated vaccine strains SG9R, specifically the Dtx-9RL and Dtx-9RM strains lacking lpxL or lpxM genes to attenuate virulence. The manuscript is very well written and are experimentally sound. In the introduction, the authors indicates that penta-acylated LPS of lpxL and lpxM mutants, but not double mutant, expresses reduced toxicity of LPS itself. I am wondering if authors had any additional data they can incorporate showing the effect of double mutant strain in the virulence?

- We appreciate your kind words. We have made lpxM and pagP double mutant but found no difference in in vitro results. We have considered making the lpxL and lpxM double mutant, but as demonstrated in the manuscript, Dtx-9RL was over-attenuated, resulting in low efficacy as for live vaccine. Therefore, it is speculated that the double mutants of lpxL and lpxM will be over-attenuated as well.

  • Minor comment: Perhaps Table 1 can go into supplementary section to improve readability.

- We agree to the reviewer’s comment. We moved Table 1 into the supplementary section.

Reviewer 3 Report

Overall, this is a clear, concise, and well-written manuscript. The introduction is relevant and theory based. Sufficient information about the previous study findings is presented for readers to follow the present study procedures. The methods are generally appropriate. Overall, the results are clear and the figures are pretty good. Overall, this is a high quality manuscript that developed optimal vaccine strains that were safe and highly protective even in young chicks, and they might be useful for providing protection against FT as well as paratyphoid caused by SE in poultry.

This paper brings attention to the ongoing problem of optimized detoxification of a live attenuated vaccine strain (SG9R) to improve vaccine strategy against fowl typhoid. This is an important issue that needs to be discussed in the community and dealt with and this article will help push towards correcting the problem in the field.

This is a clear, concise, and well-written manuscript. The introduction is relevant and theory based. Sufficient information about the previous study findings is presented for readers to follow the present study procedures. The methods are generally appropriate. The results are clear and the figures are pretty good. The conclusions consistent with the evidence and arguments presented, they address the main question posed.

In this study, the author developed Safe-9R by knocking out the rfaJ gene of SG9R and examined the efficacy and toxicity of its live and killed vaccines.So, this is a high quality manuscript that developed optimal vaccine strains that were safe and highly protective even in young chicks, and they might be useful for providing protection against FT as well as paratyphoid caused by SE in poultry. However ,we don’t know the Supplementary Materials, if the author could provide them would be better.

Author Response

  • In this study, the author developed Safe-9R by knocking out the rfaJ gene of SG9R and examined the efficacy and toxicity of its live and killed vaccines. So, this is a high quality manuscript that developed optimal vaccine strains that were safe and highly protective even in young chicks, and they might be useful for providing protection against FT as well as paratyphoid caused by SE in poultry. However ,we don’t know the Supplementary Materials, if the author could provide them would be better.

- We appreciate your kind words. You can download the Supplementary Materials from the journal website.